# Selenium Nanoparticles Based on *Morinda officinalis* Polysaccharides: Characterization, Anti-Cancer Activities, and Immune-Enhancing Activities Evaluation In Vitro

**DOI:** 10.3390/molecules28062426

**Published:** 2023-03-07

**Authors:** Mengxin Yao, Yuan Deng, Zhimin Zhao, Depo Yang, Guohui Wan, Xinjun Xu

**Affiliations:** School of Pharmaceutical Sciences, Sun Yat-sen University, Guangzhou 510006, China

**Keywords:** selenium nanoparticles, *Morinda officinalis* polysaccharides, anti-cancer activities, immune enhancement

## Abstract

Recently, selenium nanoparticles have been drawing attention worldwide, and it is crucial to increase the stability of nano-Se. *Morinda officinalis* polysaccharides (MOP) are the main active component in *Morinda officinalis* radix. However, their low activity has limited their application. A novel selenium nanoparticle (Se-MOP) was prepared to solve these problems using MOP as a dispersant. The zeta potential was measured to evaluate the stability, and UV and ATR-FTIR were used to investigate the binding type of selenium and MOP. The morphology was observed by the TEM method. Furthermore, the inhibitory effect on five selected cancer cells (HepG2, MCF-7, AGS, PC9, and HCT8) was evaluated, showing remarkable inhibition of all five cancer cells. The mechanism of inhibition was also investigated by cell circle assay, and it was found that Se-MOP could induce cell circle G0/G1 phase arrest. Immune-enhancing activities were evaluated by measuring the proliferation and cytokines of mouse spleen lymphocytes in vitro and quantitative RT-PCR. The results indicated that single stimulation of Se-MOP and synergistic stimulation with PHA or LPS increased immune capacity and improved immune by increasing the expression of cytokines.

## 1. Introduction

Selenium (Se) is an essential element for human beings. It was discovered in nature in two forms: inorganic and organic selenium. Sodium selenite and selenate are two forms of inorganic selenium that are included as components of dietary supplements [1]. Selenoproteins and selenium analogs of amid acids (e.g., selenomethionine and selenocysteine) are two forms of organic selenium found in plants, animal foods, and selenium yeast [2,3]. In metabolism, selenium plays a vital role in synthesizing the active center of glutathione peroxidase and other significant enzymes. In low selenium supply, the specific proliferation ability of T cells and the number of natural killer cells are both decreased [1,4]. The risk of suffering from Keshan disease [5], cardiovascular diseases [6,7], type 2 diabetes [8], and cancer [9,10,11] may rise due to a deficiency of selenium.

China has abundant selenium resources but needs to be more balanced. A long and narrow selenium-deficiency zone from the east to the southwest hinterland has caused over one hundred million people to suffer from low selenium supply [12]. However, Enshi, a city in Hubei Province, China, is renowned as the “Selenium Capital” worldwide. Several incidents of selenium poisoning occurred in this city due to excess inorganic selenium supplements in the soil [13]. To fully use selenium resources in China, reducing the toxicity of inorganic selenium and improving the efficiency of selenium supplementation are crucial.

Nano-selenium is a new zero-valent elemental selenium type, the least toxic form of selenium known [14]. Usually, zero-valent elemental selenium cannot be absorbed by humans directly. However, nano-selenium has completely different properties, including color changing (from black or gray to orange), increased water solubility, and significantly increased bioactivities [15]. It was reported in one study that the nano-selenium particle size and bioactivities were negatively correlated [16]. The biggest problem with selenium nanoparticles is their tendency to aggregate [17]. Based on this situation, researchers sought a proper dispersant and turned their attention to polysaccharides, a naturally macromolecular substance. Liu et al. reported a redox reaction between sodium selenite and ascorbic acid generated nano-selenium through the addition of *Oudemansiella radicata* polysaccharides as a dispersant [18].

*Morinda officinalis* radix, a geo-authentic crude drug of Guangdong province, is a widely-used traditional Chinese medicine. *Morinda officinalis* polysaccharide (MOP) is one of the foremost effective components with many pharmacological activities, such as antioxidation, anti-depression, anti-osteoporosis, immune system enhancement, and reproductive capacity enhancement [19,20,21,22,23,24]. Xu obtained a homogeneous MOP, and its molecular weight was 63 KDa. It comprised arabinose, glucose, rhamnose, galactose, etc. [25]. Moreover, MOP has been used as an immunopotentiator [26] and leukopenia treatment in mice [27]. Furthermore, the MOP has numerous active hydroxyl groups and excellent water solubility with non-toxicity. These properties indicate the potential of MOP becoming a suitable nanoparticle dispersant. Therefore, in this article, a novel nano-selenium-MOP (Se-MOP) was prepared from sodium selenite and ascorbic acid and used MOP as a dispersant. The characterization of Se-MOP was determined by various methods. Furthermore, five kinds of cancer cells were chosen to evaluate the inhibitory effect. Moreover, the immune enhancement was also evaluated in vitro.

## 2. Results

### 2.1. Characterization of Se-MOP

#### 2.1.1. Measurement of Size and Zeta-Potential

The results are shown in Table 1. The particle size of Se-MOP was 67.25 ± 0.99 nm (*n* = 3). In contrast, the particle size of MOP was 153.97 ± 10.65 nm (*n* = 3). The polydispersity index (PDI) average of Se-MOP was 0.10 ± 0.02 (*n* = 3). All PDI values were close to 0.1, manifesting a narrow range of distribution (Figure 1), suggesting the even size of all particles. The zeta potential of Se-MOP was −21.17 ± 0.47 mV (*n* = 3). The zeta potential result showed that selenium nanoparticles with MOP had good stability. The detailed size distribution report by intensity is shown in Appendix A.

#### 2.1.2. UV Spectrometry Analysis

The result of UV spectrometry analysis is shown in Figure 2. MOP did not have absorption peaks in the 200–800 nm wavelength range. However, sodium selenite had a maximum absorption peak of 212 nm, while Se-MOP had a maximum absorption peak of 266 nm. The different absorption of UV spectrometry analysis results demonstrated that Se-MOP varied from MOP and sodium selenite.

#### 2.1.3. ATR-FTIR Spectrometry Analysis

The typical characteristic peaks of Se-MOP and MOP are shown in Figure 3. Hydroxyl stretching vibration peaks were at 3238 cm^−1^ and 3257 cm^−1^. Characteristic peaks at 1591 cm^−1^ and 1594 cm^−1^ were related to binding water. CH_2_ shear vibration peaks were at 1405 cm^−1^ and 1404 cm^−1^. The pyran ring’s C-H variable angle vibration peaks were at 1400 cm^−1^~1200 cm^−1^. C-O-C stretching vibration peaks of the pyran ring were at 1026 cm^−1^ and 1022 cm^−1^. Asymmetric ring stretching vibration peaks of D-glucopyranose were at 929 cm^−1^ and 934 cm^−1^. Transverse vibration peaks of the methine groups were at 871 cm^−1^ and 876 cm^−1^. The furan ring’s C-H variable angle vibration peaks were at 829 cm^−1^ and 818 cm^−1^. Moreover, when MOP was combined with selenium, hydroxyl vibration peaks shifted from 3257 cm^−1^ to 3238 cm^−1^, and a new peak appeared at 668 cm^−1^.

#### 2.1.4. Transmission Electron Microscopy (TEM) Analysis

Different views of Se-MOP were observed by TEM (Figure 4). The particle size of Se-MOP was 40–80 nm. Furthermore, these particles were spherical, even-sized, and well-dispersed. TEM analysis attested that MOP was a good dispersant for nano-se.

### 2.2. Selenium Content Determination

The plot was drawn with selenium content (μg) as abscissa and absorbance value as ordinate. The standard equation was y = 0.251x + 0.0089, r = 0.9999. The selenium content of Se-MOP was (0.99 ± 0.01)% (*n* = 3).

### 2.3. Results of Inhibitory Effect on Different Types of Cancer Cells

#### 2.3.1. Anti-Tumor Activity of Se-MOP and MOP

The in vitro anti-tumor activity of Se-MOP and MOP on HepG2, MCF-7, AGS, PC9, and HCT8 cells was evaluated using CCK8 assay. As shown in Figure 5, after treated with different concentrations of Se-MOP and MOP (0, 3.125, 6.25, 12.5, 25, 50, 100, 200, and 400 μg/mL) for 48 h and 72 h, Se-MOP showed significant inhibition effects on HepG2, MCF-7, AGS, PC9, and HCT8 cells (*p* < 0.01) and PC9, MCF-7,and AGS cells were inhibited in a dose-dependent manner as well. At the same time, MOP only had weak anti-tumor activity. Among these cancer cells, Se-MOP must have shown the most significant inhibition effect by HepG2, with the lowest IC50 value of 2 μg/mL at 48 h and 1 μg/mL at 72 h.

#### 2.3.2. Cell Cycle Arrest Induced by Se-MOP

A PI staining assay was performed and analyzed using flow cytometry. After being treated with different concentrations of Se-MOP (0, 5, 25, 50, 100, 200, and 400 μg/mL) for 24 h, the cell percentage of G1 phases was decreased, while the percentage of G2 phases was increased in a dose-dependent manner (Figure 6A). These cancer cells phases were further quantified by Modfit software. The percentage of cancer cells in the G1 region was decreased from 53.43% to 4.328% after treatment with Se-MOP (0, 5, 25, 50, 100, 200, and 400 μg/mL) for 24 h (*p* < 0.05, *p* < 0.01, *p* < 0.001, and *p* < 0.0001) (Figure 6B). Meanwhile, the percentage of cancer cells in the G2 region was increased from 15.35% to 64.87% after being treated with Se-MOP (0, 5, 25, 50, 100, 200, and 400 μg/mL) for 24 h (*p* < 0.05, *p* < 0.01, *p* < 0.001, and *p* < 0.0001) (Figure 6B).

### 2.4. Immune Enhancing Activities Assay

#### 2.4.1. Effects of Se-MOP and MOP on Lymphocyte Proliferation and Maximum Proliferation Rate

With single stimulation of Se-MOP, or in synergistic stimulation of Se-MOP (0.195–200 μg/mL) with PHA (10 μg/mL) and LPS (5 μg/mL) after 48 h, lymphocyte cells were detected for the viability using CCK8 assay. As shown in Figure 7A, after being treated with Se-MOP at 0.195–200 μg/mL, compared to the control group, the number of lymphocytes increased at 3.125–25 μg/mL, in which the maximum proliferation rate was at 12.5 μg/mL. At the same time, inhibition was also shown at other concentrations. Similar results were displayed in synergistic stimulation of Se-MOP (0.195–200 μg/mL) with PHA (10 μg/mL) and LPS (5 μg/mL) after 48 h. The concentration of Se-MOP from 0.195 μg/mL to 12.5 μg/mL revealed that lymphocyte proliferation was in dose dependence except for 12.5 μg/mL. For the LPS treatment with Se-MOP, there were an increased number of lymphocytes at 1.56–25 μg/mL, and the concentration of maximum proliferation rate was also 12.5 μg/mL. These results led to a similar conclusion: Se-MOP promoted lymphocyte proliferation and increased immunity.

#### 2.4.2. Effects of Se-MOP on Gene Expression of IL-2, IL-4, and IFN-γ in Mouse Spleen Lymphocytes

To avoid the cytotoxicity of lymphocytes, Se-MOP at the indicated concentrations (0.195, 0.39, 0.78, 1.56, 3.125, 6.25, 12.5, 25 μg/mL) was verified for safety and at proliferation concentration for the following assays. The different concentrations of Se-MOP (0.195, 0.39, 0.78, 1.56, 3.125, 6.25, 12.5, 25 μg/mL) were added after being treated with PHA (10 μg/mL), respectively, for 48 h and then the mRNA expressions of cytokines were detected. The relative expression of IFN-γ, IL-2, and IL-4 mRNA in each group is shown in Figure 7B and Table 2. Compared to the PHA control group, the Se-MOP concentration at 0.78–6.25 μg/mL showed significantly increased relative expression of IFN-γ, at 3.924, 5.313, 3.872 and 2.676 times higher, respectively. Additionally, at the Se-MOP concentration of 0.195–25 μg/mL, relative expressions of IL-2 were 1.492, 1.677, 2.538, 3.754, 2.21, 1.849, 1.625, and 1.496 times higher than that of the PHA group, among which at the concentration of 1.56 μg/mL, the content of IL-2 was the highest. Moreover, it was a similar increased content at the concentration of 0.39–6.25 μg/mL, with 1.721, 7.418, 2.09, 1.948, and 1.764 times higher than that of control group. Se-MOP could improve immunity by increasing cytokine expression, such as that of IFN-γ, IL-2, and IL-4.

## 3. Discussion

Zeta potential is a significant indicator in evaluating the stability of nanoparticles. Zeta potential, which refers to the potential of the shear plane, is an important index to characterize the stability of a colloidal dispersion system and an index to measure the intensity of mutual repulsion or attraction between particles. The positive or negative of the zeta potential represents which charge the particle carries. The absolute value of the zeta potential is positively correlated with the repulsion between particles. The higher the absolute value, the more stable the dispersion system. When there is no noticeable difference in the zeta potential, the smaller the particle size, the higher the solution’s stability. In Section 2.1.1, MOP mean size was trending by 12.5%. Moreover, the count rate and size increased, suggesting the sample may have been aggregating. After the modification of MOP, the two influence each other, the particle size of Se-MOP decreased, the zeta potential exceeded the spatial stability value −20 mV, and to the sample did not aggregate easily. It was reported that the zeta potential should be close to 20 mV to maintain sterically stable [28]. Gao et al. determined that the zeta potential of Polyporus umbellatus polysaccharide nano-selenium was −22.6 mV, and it could be stored for 84 days at 4 °C in the dark [29].

The color of nano-selenium changes obviously in the process of formation. During the formation, the color of the nano-selenium solution changes from colorless to red, leading to surface plasmon resonance [30]. The surface plasmon resonance of metal is essential in determining metal nanoparticles’ optical properties. Macroscopically, this resonance is shown as the light absorption of metal nanoparticles, which have absorption bands in the UV-Vis region [31]. It was reported that the position of the maximum absorption peak is related to the particle size of nano-selenium in UV detection. When the particle size was about 200 nm, the maximum absorption peak was near 600 nm; when the particle size was lower than 100 nm, the maximum absorption peak was 200–300 nm [32]. Therefore, these results confirmed that new, stable particles were formed, and the particle size was below 100 nm.

ATR-FTIR was used to investigate the binding type preliminarily between selenium nanoparticles and MOP. Apparently, Se-MOP had the skeleton of MOP. Moreover, the shift of the hydroxyl vibration peaks to a lower wavenumber and increased intensity indicated that hydrogen bonds were formed. More intriguingly, there was a weak peak of Se-MOP at 668 cm^−1^, which could be considered an Se-H bond absorption peak. Zhu et al. reported the same peak at 669 cm^−1^ characterized by FT-IR, and further proved there was Se-H in C-6 connections using NMR spectra [33]. Results from ATR-FTIR proved that Se-MOP had the skeleton of MOP, and selenium may combine with MOP through Se-H bonds. The latter result requires further evidence to be obtained using NMR spectra in follow-up studies.

TEM can directly observe the size and morphology of nano-particles, which is the first choice of many researchers [34]. The results discussed in Section 2.1.4 showed that the Se-MOP particles were spherical, even-sized, and well-dispersed. The TEM, size, and zeta potential, and UV-Vis results confirmed each other. Moreover, it was found in Section 2.2 that the selenium content of Se-MOP was high. The World Health Organization (WHO) ‘s maximum allowable daily selenium intake is 400 μg [35]. If Se-MOP were used to supplement selenium, no more than 0.04 g per day would be required, which would significantly improve the efficiency of the selenium supplement.

The anti-tumor activity result provided evidence that the new formation of Se-MOP could significantly improve the anti-tumor activity of native MOP in several tumor cells, which has also been verified by previous studies. Zhang et al. confirmed that selenium nanoparticles from dandelion polysaccharide exerted anti-tumor activity on A549 cells, HepG2 cells, and Hela cells through inducing cell apoptosis [36]. Similarly to this conclusion, rabinogalactans/selenium nanoparticles composites (LAG-SeNPs) constructed by Tang et al. exerted a better inhibition of A549 cell, HepG2 cell, and MCF-7 cell proliferation [37]. A cell cycle assay was performed and analyzed using flow cytometry to investigate further the possible anti-tumor mechanism of Se-MOP on HepG2 cells. As a process of cell proliferation, the cell cycle is an orderly occurrence of the replication of genomic DNA, followed by an equal division of the genome into two similar cells, initially divided into two phases: mitosis and interphase. Whether cells complete the proliferation process is closely related to whether cells experience the whole phase of the cell cycle smoothly, which is regulated with extreme precision. The different stage of the cell cycle has different characteristics and functions of regulation. For interphase, it was further divided into three phases: G1 phase, S phase, and G2 phase. The G1 phase is characterized by the synthesis of mRNAs and proteins required for DNA replication. At the same time, the S phase is the phase of DNA replication. The G2 phase is the late stage of DNA synthesis and the preparation period for mitosis. With fewer and fewer mRNAs and proteins synthesized, it was apparent that there was not sufficient material to support DNA replication, which also caused cell cycle arrest. Together, the cell cycle assay results demonstrated that Se-MOP could inhibit the proliferation of cancer cells by cell cycle G0/G1 phase arrest. Coincidentally, Venkateswaran [38] also reported similar results for human PCA cells (LNCaP, PC3, PC3-AR2, and PC3-M) incubated with and without Seleno-DL-methionine, in which treatment with selenium caused G1 arrest and an 80% reduction in the S phase of LNCaP [37]. In this study, we investigated the preliminary mechanisms underlying the short-term (24 h) effects of Se-MOP on the cell cycle of liver cancer cells. The results of cell cycle arrest may lead to apoptosis [39]. Previous studies have shown that different methods can induce apoptosis in HepG2 cells. It has been reported that selenium–sorafenib complexes can effectively activate the Ca^2+^ signaling system and cause endoplasmic reticulum stress [40], while glucan-selenium nanoparticles (Glucan-SeNPs) can activate Caspase-3 and Caspase-9 [41]. This experiment provides a basis for exploring the specific mechanism of apoptosis in the future.

The previous results in Section 2.4.1 showed that Se-MOP stimulated lymphocyte proliferation in vitro and enhanced cellular immune function. The mouse spleen lymphocytes were conducted as the experimental model to explore the mechanism of enhancing the immunity of selenium. After being stimulated by PHA, the lymphocytes were transformed into lymphoblasts, further proliferated, and released lymphokines, increasing the phagocytosis of macrophages. Lipopolysaccharide (LPS) is a potent activator of immune cells, including B cells, monocytes, and macrophages, which is required for stimulation to produce cytokines. Cytokine is a small, non-specific immune molecule other than immunoglobulin and complement, which can mediate and regulate the immune and inflammatory responses of the body. They have a variety of biological activities, such as regulating immunity, inhibiting tumor proliferation, and so on. IFN-γ and IL-2 play essential roles in the body’s immune response and are natural immune enhancers, while IL-4 mainly is involved in maintaining the normal immune function of the organism. In general, the immune ability was enhanced by single stimulation of Se-MOP and synergistic stimulation with PHA or LPS, increasing the expression of cytokines, such as IFN-γ, IL-2, and IL-4. From the results of the cytokine expression, when the concentration of Se-MOP was moderate, the relative expression of cytokines was the highest. When the concentrations of Se-MOP were 1.56 μg/mL, 1.56 μg/mL, and 0.78 μg/mL, the relative expression levels of IFN-γ, IL-2, and IL-4 cytokines were the highest. This is consistent with the trend of PHA synergistically stimulating mouse spleen lymphocytes. Mao et al. [42] found that the combination of Astragalus polysaccharide and selenium nanoparticles (Se-GP11) had no toxic effect on HepG-2 cells in vitro, but could significantly inhibit the growth of HepG2 tumors in vivo. It increased the level of interleukin 2. This is similar to the results of our in vitro experiment, providing a research direction for subsequent in vivo experiments. In previous studies, selenium can synergize with sufficient IL-4 to activate the receptor γ (PPAR-γ)-reliant pathway, promoting macrophages from M1 to M2 [43]. In this experiment, Se-MOP can increase the relative expression of IL-4. Combined with the previous studies, the effect of Se-MOP may also be related to the inhibition of inflammation in M2 macrophages, which provides a research direction for later studies. IFN-γ is associated with CD8+ T cell function [44]. In future studies, the cytotoxic activity of selenium on tumors and its effects on the immune microenvironment of tumors can be studied.

This article aims to investigate the mutual influence between nano-selenium and MOP, as well as the improvement of MOP activity by selenium. The possible binding modes between nano-selenium particles and MOP are speculated in this article and require further verification using one-dimensional or two-dimensional NMR methods. Furthermore, the activity of bare nano-selenium particles will be compared with Se-MOPs in subsequent experiments. Moreover, in vivo experiments are required to verify the in vitro activity in this article. Finally, further research on the mechanism of anti-cancer and immune enhancement is needed.

## 4. Materials and Methods

### 4.1. Materials and Reagents

The MOP was prepared as reported previously [45]. Sodium selenite was purchased from Sigma-Aldrich (St. Louis, MI, USA). Ascorbic acid was purchased from Zhiyuan Co. (Tianjin, China). Carbon support films were purchased from Zhongjingkeyi (Beijing, China) Films Technology Co., Ltd. (Beijing, China). Phytohemagglutinin (PHA) was purchased from Selleck Co., Ltd. (Huston, TX, USA). Lipopolysaccharide (LPS) was purchased from Sigma-Aldrich (St. Louis, MI, USA). Trizol reagent was purchased from ThermoFisher (Waltham, MA, USA). Cell Counting Kit-8 was purchased from GlpBio (Montclair, NJ, USA). Evo M-MLV RT Premix for qPCR was purchased from Accurate Biology (Changsha, China). SYBR Green-Premix Pro Taq HS Qpcr Kit II was purchased from Accurate Biology (Changsha, China). Mycoplasma Stain Assay Kit was purchased from Beyotime (Shanghai, China). All other reagents were analytically pure.

### 4.2. Apparatus

The Supo HHS-6 electro-thermostatic water bath and the Evela N-1200A rotary evaporator were used. The freeze-drying method was operated by Biosafer Biosafer-10A and Christ Alpha 1–2 LD plus vacuum freeze-dryers. The size distribution and zeta potential were determined using a Malvern Zetasizer nano ZS analyzer. The UV used a Shimadzu UV-2600 UV-vis spectrophotometer, and the ATR-FTIR used a Perkin Elmer Spectrum two infrared analyzer. The JEOL JEM-2010HR was operated for TEM observation.

The cell cycles were performed by a Beckman CytoFLEX S flow cytometry analyzer. Cell proliferation was determined with a Multiskan FC ThermoFisher microplate reader. The quantitative RT-PCR was conducted with 7500 apparatus, Applied Biosystems (Waltham, MA, USA).

### 4.3. Preparation of Se-MOP

MOP was dissolved in water to prepare a 5 mg/mL solution. Sodium selenite was dissolved in water to prepare a 10 mmol/L solution. Ascorbic acid was dissolved in water to prepare a 50 mmol/L solution and kept in the dark. MOP solution of 2 mL and sodium selenite solution of 5 mL was put in a 50 mL colorimetric tube. Water was added to the 25 mL scale line and mixed well. The mixture was pre-heated in the water bath 50 °C for 20 min, and an ascorbic acid solution of 3 mL was added to the mixture (molar ratio of sodium selenite to ascorbic acid = 1:3). Water was added to the 50 mL scale line and mixed well. The mixture was heated in the water bath at 50 °C for 4 h. The reaction was terminated by stopping heating. The reaction mixture was transferred into a round-bottom flask and concentrated by rotary evaporation. The concentration was vacuum freeze-dried for 24 h. Eventually, the Se-MOP dry powder was collected.

### 4.4. Characterization of Se-MOP

#### 4.4.1. Measurement of Particle Size and Zeta Potential

Se-MOP solution in a quantity of 10 mL before freeze-drying and diluted MOP solution from Section 4.3 were measured for particle size, PDI, and zeta potential using a laser particle size analyzer.

#### 4.4.2. UV Spectrometry Analysis

Se-MOP dry powder was dissolved in water to prepare a 0.3 mg/mL solution. Sodium selenite and MOP were prepared in the same concentrations in separate solutions by the same method. Each solution was scanned by an Ultraviolet spectrophotometer with a quartz cuvette in the 200–800 nm wavelength range, using water as reference.

#### 4.4.3. ATR-FTIR Spectrometry Analysis

Se-MOP dry powder 10 mg was placed on the prism and pressed into a thin tablet. The tablet was measured on an ATR-FTIR spectrometer in the range of 4000–400 cm^−1^.

#### 4.4.4. TEM Analysis

Before freeze-drying, Se-MOP solution 5 mL was processed by ultrasonic machine for 20 min. Afterward, the solution was filtrated into a syringe through the 220 μm millipore filter. The filtrate was dropped into a carbon support film (mesh number: 300). The carbon support film was placed in a dryer for 72 h, and then the morphology of the Se-MOP was observed by TEM.

### 4.5. Selenium Content Determination

The method was derived from [46,47] and was modified. Organic matter was removed from the sample after digestion. The zero-valent nano-selenium was oxidized to Se (IV) by the oxidant. O-phenylenediamine reacted with Se (IV) to form a complex (Figure 8). The complex was extracted by toluene. A UV-vis spectrophotometer detected the content of the complex. Sodium selenite was used as the reference material to draw the standard curve. The detailed procedures are demonstrated as follows.

#### 4.5.1. Standard Curve Plotting

All procedures were operated in the dark. Sodium selenite 10.27 mg was dissolved in water in a 25 mL volumetric flask. Water was precisely added to the scale line to get the standard selenium solution (selenium content: 187.56 μg/mL). Standard selenium solution (0.1 mL, 0.2 mL, 0.4 mL, 0.8 mL, and 1.6 mL) was separately transferred into a 20 mL volumetric flask. Each volumetric flask was added to 2 mL 2% o-phenylenediamine solution and 2 mL of water. Later, the pH value of the reaction mixture was adjusted to two. Water was added to the scale line, and the system reacted for 1 h. After the reaction, the mixture was extracted with 20 mL toluene, and the organic layer was retained. The absorbance of the organic layer was measured at a wavelength of 334 nm by an ultraviolet spectrophotometer with a quartz cuvette, using toluene as a reference. A standard curve was plotted based on the former results.

#### 4.5.2. Se-MOP Digestion and Determination

Se-MOP dry powder in a quantity of 50 mg was placed in a ceramic crucible with a lid. Concentrated nitric acid in a quantity of 1 mL was added to the Se-MOP solution, which was then digested for 8 h. Afterwards, the crucible was heated with an electric hot plate until the liquid within was nearly dried. H_2_O_2_ solution (30%) of 0.3 mL was added dropwise. The crucible was heated again until the liquid within was colorless. The inner side of the crucible was washed with 0.1 mL water, and the crucible was heated until no smoke was produced. The digested liquid was transferred into a 10 mL volumetric flask, and water was precisely added to the scale line. The sample was made for later selenium determination and a 2 mL sample was transferred into a 20 mL volumetric flask. Other procedures were the same as described in Section 4.5.1. The absorbance was used to calculate the selenium content.

### 4.6. Inhibitory Effect on Different Types of Cancer Cells

#### 4.6.1. Cancer Cell Culture

HCT8, AGS, HePG2, MCF-7, and PC9 cell lines were obtained from the American Type Culture Collection (ATCC). HCT8 and PC9 cells were cultured in Roswell Park Memorial Institute-1640 (RPMI1640, Corning, USA) with 10% FBS and 1% Penicillin and Streptomycin at 37 °C in 5% CO_2_. AGS, HePG2, and MCF-7 cells were cultured in Dulbecco’s modified Eagle’s medium (DMEM, Corning, USA), with 10% FBS and 1% Penicillin and Streptomycin at 37 °C in 5% CO_2_. All cell lines were stored in multiple backups upon receipt to reduce risk of phenotypic drift, and tested to determine whether they were mycoplasma-free by Mycoplasma Stain Assay Kit.

#### 4.6.2. Cell Proliferation

For the CCK8 assay, HCT8, AGS, HePG2, MCF-7, and PC9 cells were seeded at 5000 cells per well in 96-well plates with fresh medium overnight. Se-MOP and MOP were diluted with serum-free medium to a final concentration of 0 μg/mL, 3.125 μg/mL, 6.25 μg/mL, 12.5 μg/mL, 25 μg/mL, 50 μg/mL, 100 μg/mL, 200 μg/mL, and 400 μg/mL, respectively, and added into 96-well plates in the next day. Cell viability was assayed by using the Cell Counting Kit-8 at 48, 72 h. The microplates were incubated at 37 °C for an additional 2 h. Absorbance was read at 450 nm using a microplate reader (Multiskan FC, ThermoFisher, USA).

#### 4.6.3. Cell Cycle Assays

Propidium iodide staining and flow cytometry were adapted from Cell Cycle Staining Kit. HePG2 cells were plated in 60 mm cell culture dishes overnight for cell cycle detection. Se-MOP was diluted with serum-free medium to final concentrations of 5 μg/mL, 25 μg/mL, 50 μg/mL, 100 μg/mL, 200 μg/mL, and 400 μg/mL, and added into cell culture dishes for another 24 h. These cells were trypsinized, collected, and washed twice with PBS. They were then resuspended in 1 mL DNA staining solution and 10 μL permeabilization solution, oscillated for 5–10 s, and incubated at 37 °C for 30 min in the dark. Then, cells were stained, followed by flow cytometry analysis.

### 4.7. Immune Enhancing Activities Assay

#### 4.7.1. Determination of Proliferation of Mouse Spleen Lymphocytes In Vitro

All mouse procedures were approved and carried out in accordance with the Institutional Animal Care and Use Committee of Sun Yat-Sen University (No. 44008500029369). We used 5-week-old ICR male mice for all our mouse experiments. The method of preparing mouse spleen lymphocytes followed that of Luo [48] and they were counted by trypan blue staining. Se-MOP and MOP were diluted with RPMI1640 medium from 200 μg/mL to 0.195 μg/mL in 11 concentration gradients (200, 100, 50, 25, 12.5, 6.25, 3.125, 1.56, 0.78, 0.39, and 0.195 μg/mL). Mouse spleen lymphocytes were seeded at 150,000 cells/mL at 100 μL per well in 96-well plates with fresh RPMI1640 medium containing 10% FBS, 1% penicillin and streptomycin. PHA solution (to a final concentration of 10 μg/mL), LPS solution (to a final concentration of 5 μg/mL), and the blank medium were added and incubated at 37 °C, respectively. Then, Se-MOP and MOP dilution for each concentration were added into 96-well plates, respectively, and left overnight.

Cell viability was assayed using Cell Counting Kit-8 over 48 h. The microplates were incubated at 37 °C for an additional 2 h. Absorbance was read at 450 nm using a microplate reader.

#### 4.7.2. Determination of Cytokines of Mouse Spleen lymphocytes In Vitro and Quantitative RT-PCR

The spleen lymphocytes of 5-week-old ICR male mice was prepared as above. Mouse spleen lymphocytes were seeded at 1 × 10^7^ cells/mL at 1 mL per well in 6-well plates with fresh RPMI1640 medium containing 10% FBS, 1% penicillin and streptomycin and PHA solution (to a final concentration of 10 μg/mL) was added and incubated at 37 °C. Se-MOP was diluted with RPMI1640 medium from 200 μg/mL to 0.195 μg/mL in eleven concentration gradients (25, 12.5, 6.25, 3.125, 1.56, 0.78, 0.39, and 0.195 μg/mL) and added to 6-well plates, respectively. The spleen lymphocytes were collected for 48 h incubation.

Total RNA was isolated using Trizol reagent according to the manufacturer’s instruction, and cDNA was synthesized using the Evo M-MLV RT Premix for qPCR. The resulting cDNA was used for quantitative RT-PCR using SYBR Green-Premix Pro Taq HS Qpcr Kit II in 7500 apparatus (Applied Biosystems). β-actin mRNA was the housekeeping gene used to normalize the expression of mRNAs. RT-PCR primer sequences are listed in Appendix A.

## 5. Conclusions

A new kind of selenium nanoparticle, Se-MOP, was prepared successfully. The UV spectra demonstrated that new nanoparticles existed. The TEM results showed that the Se-MOP particles were spherical, even-sized, and well-dispersed, and the zeta potential proved its good stability. Additionally, the binding type between selenium particles and MOP was tentatively investigated by ATR-FTIR spectra. Se-MOP showed significant inhibition effects on all selected five cancer cells. The mechanism may be that Se-MOP could inhibit the proliferation of cancer cells by cell cycle G0/G1 phase arrest. Single Se-MOP stimulation and synergistic stimulation with PHA or LPS increased immune capacity. When the concentration of Se-MOP was in the middle (1.56, 3.125, 6.25, 12.5, 25 μg/mL), the cytokines’ relative expression was generally significant. This was consistent with the trend of the Se-MOP and PHA synergistic stimulation results. These results indicate that Se-MOP could improve immunity by increasing cytokine expression.

However, the possible binding modes between nano-selenium particles and MOP are only speculated in this article and require further verification. The activity of bare nano-selenium particles should be compared with that of Se-MOP. Moreover, further research on anti-cancer and immune enhancement mechanisms and in vivo experiments are needed. Despite the limitations, we discovered that nano-selenium was successfully dispersed with the help of *Morinda officinalis* polysaccharide in this study. After combining with nano-selenium, the activity of *Morinda officinalis* polysaccharide was significantly enhanced. It can be concluded that polysaccharides, as the dispersants of nano-selenium, have the advantages of controllable nanoparticle morphology and a simple and safe synthesis method. Furthermore, such a combination can also improve the biological activity of polysaccharides and has excellent potential for application in the future.

## Figures and Tables

**Figure 1 molecules-28-02426-f001:**
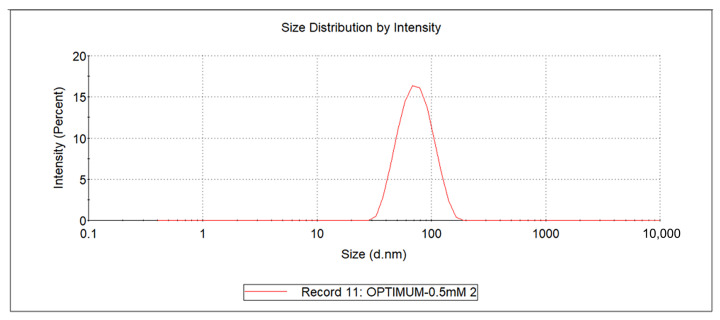
Se-MOP size distribution by intensity. The horizontal coordinate represents the particle size (nm), and the vertical coordinate represents the intensity (%). All particle sizes are concentrated in the range of one peak.

**Figure 2 molecules-28-02426-f002:**
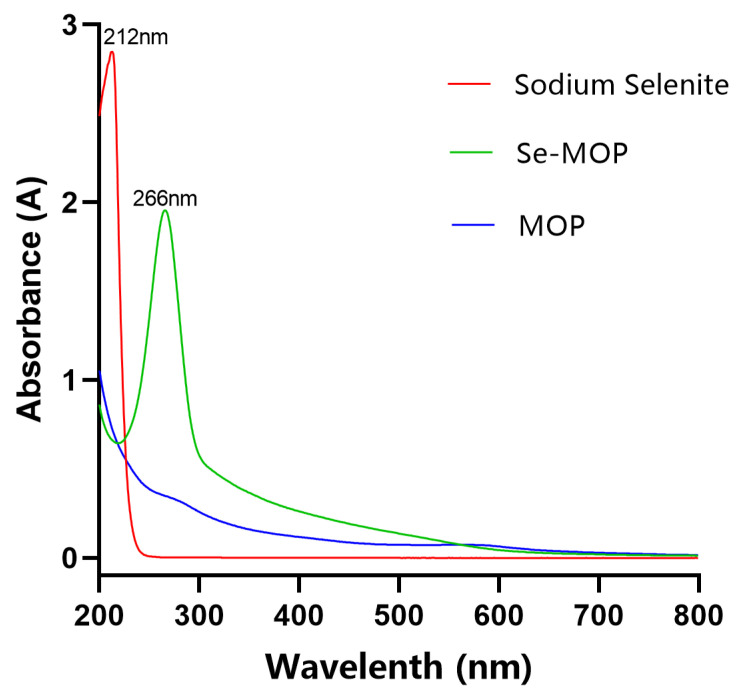
UV spectra of Se-MOP (green line), sodium selenite (red line), and MOP (blue line) in the range of 200–800 nm. The horizontal coordinate represents the wavelength (nm). The vertical coordinate represents the absorbance. The Se-MOP line peaks at 266 nm. The sodium selenite peaks at 212 nm. The MOP line does not peak.

**Figure 3 molecules-28-02426-f003:**
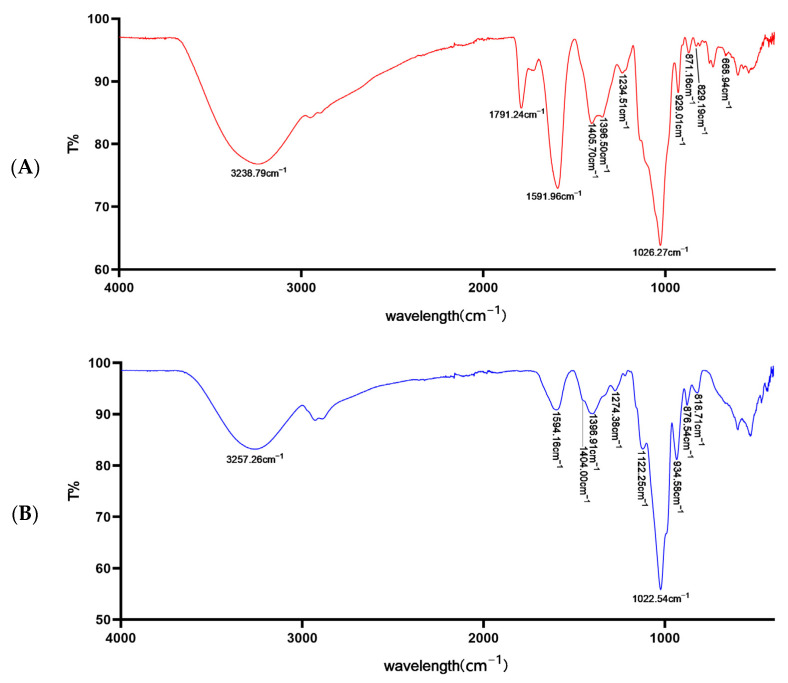
ATR-FTIR spectra of Se-MOP (**A**) and MOP (**B**) in the range of 4000–400 cm^−1^. The horizontal coordinate represents the wavelength (cm^−1^). The vertical coordinate represents the transmittance (%).

**Figure 4 molecules-28-02426-f004:**
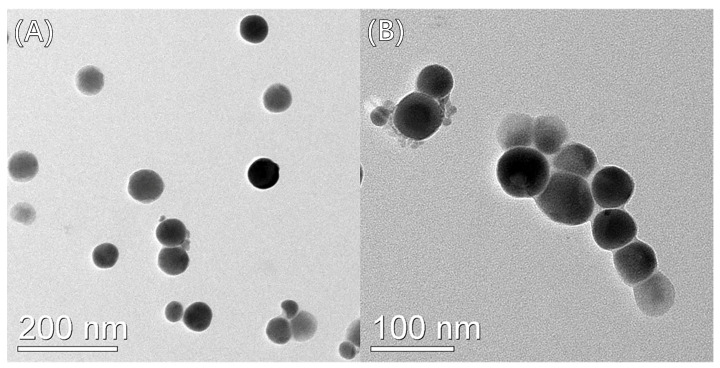
TEM photograph of Se-MOP. (**A**) Magnification: 30,000.0 × Beam: 200.0 kV. (**B**) Magnification: 50,000.0 × Beam: 200.0 kV.

**Figure 5 molecules-28-02426-f005:**
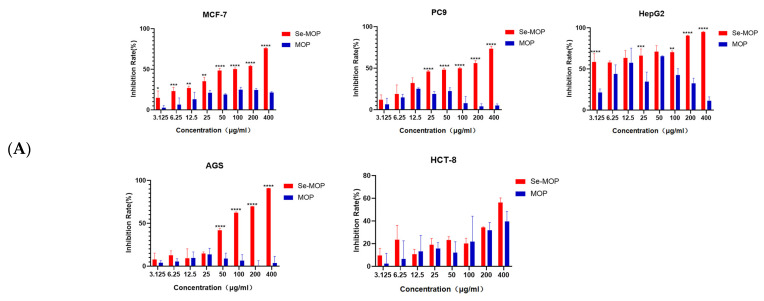
Inhibition rates of Se-MOP and MOP by CCK8 assay on human cancer cells after 48 h (**A**) and 72 h incubation (**B**). (**A**) Inhibition rate of Se-MOP and MOP (3.125, 6.25, 12.5, 25, 50, 100, 200, and 400 μg/mL) on MCF-7 cells, PC9 cells, HepG2 cells, AGS cells, and HCT-8 cells after 48 h incubation. (**B**) Inhibition rate of Se-MOP and MOP (3.125, 6.25, 12.5, 25, 50, 100, 200, and 400 μg/mL) on MCF-7 cells, PC9 cells, HepG2 cells, AGS cells, and HCT-8 cells after 72 h incubation. Data from three separate experiments are expressed as means ± SD in triplicate. * *p* < 0.05, ** *p* < 0.01, *** *p* < 0.001, **** *p* < 0.0001 vs. the control group.

**Figure 6 molecules-28-02426-f006:**
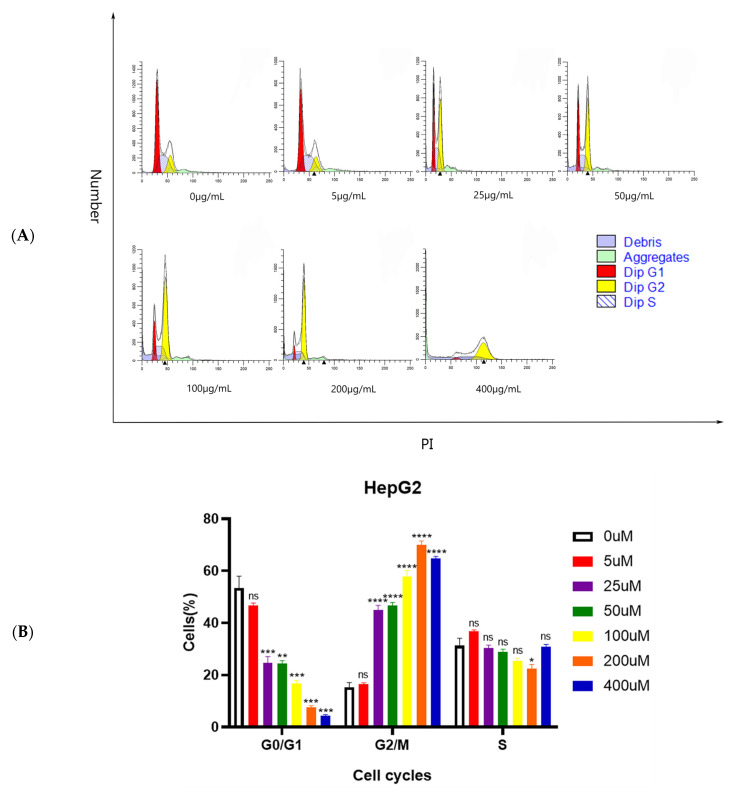
The cell cycle of HepG2 cells induced by Se-MOP (0, 5, 25, 50, 100, 200, and 400 μg/mL). (**A**) Flow cytometric analysis of HepG2 cells after being treated with different concentrations of Se-MOP (0, 5, 25, 50, 100, 200, and 400 μg/mL) for 24h. The red color region represents the G1 phase (a period from mitosis to DNA replication, which mainly synthesizes RNA and ribosomes.); the yellow color region represents the G2 phase (a late stage of DNA synthesis and a preparation period for mitosis, in which a large amount of RNA and proteins are synthesized.); slash with blue color region represents the S phase (the DNA synthesis period and histone synthesis). (**B**) Histogram of HepG2 cells percentage in a different cell cycle phase after treatment with different Se-MOP concentrations (0, 5, 25, 50, 100, 200, and 400 μg/mL) for 24 h. All values are expressed as mean ± SD at least three independent experiments. * *p* < 0.05, ***p* < 0.01, *** *p* < 0.001, **** *p* < 0.0001 vs. the control group.

**Figure 7 molecules-28-02426-f007:**
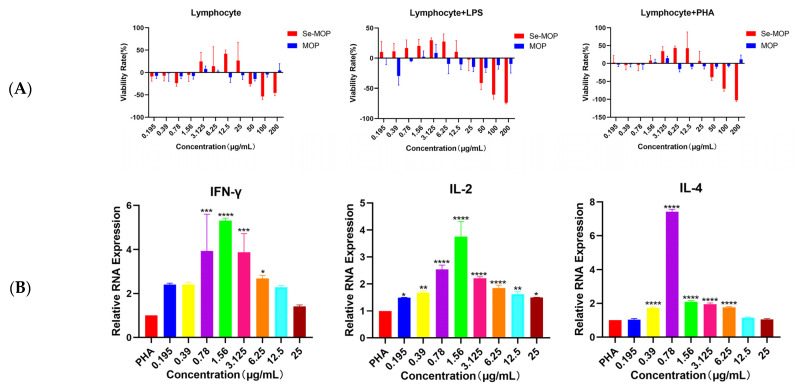
The lymphocyte proliferation of every group in single stimulation of Se-MOP or MOP and in synergistic stimulation of Se-MOP or MOP with PHA or LPS after 48 h (**A**). The changes in cytokine content of lymphocytes in each group in synergistic stimulation of Se-MOP with PHA after 48 h (**B**). (**A**) The viability rate of Se-MOP and MOP (0.195, 0.39, 0.78, 1.56, 3.125, 6.25, 12.5, 25, 50, 100, 200 μg/mL) on lymphocytes in single stimulation and in synergistic stimulation of Se-MOP or MOP with PHA (10 μg/mL), LPS (5 μg/mL) after 48 h. Data from five separate experiments are expressed as means ± SD in triplicate. (**B**) The relative RNA expression of lymphocyte IFN-γ, IL-2, and IL-4 contents in each group in synergistic stimulation of Se-MOP (0.195, 0.39, 0.78, 1.56, 3.125, 6.25, 12.5, 25 μg/mL) with PHA (10 μg/mL) after 48 h. Data from three separate experiments are expressed as means ± SD in triplicate. * *p* < 0.05, ** *p* < 0.01, *** *p* < 0.001, **** *p* < 0.0001 vs. the control group.

**Figure 8 molecules-28-02426-f008:**
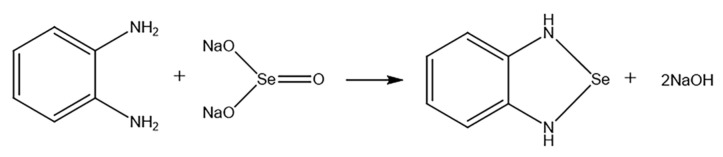
The reaction between O-phenylenediamine and Sodium Selenite.

**Table 1 molecules-28-02426-t001:** Size, PDI, and zeta potential of Se-MOP and MOP. Data from three separate experiments are expressed as means ± SD in triplicate.

Particles	Size (nm)	PDI	Zeta Potential (mV)
Se-MOP	67.25 ± 0.99	0.10 ± 0.02	−21.17 ± 0.47
MOP	153.97 ± 10.65	0.26 ± 0.01	−13.80 ± 0.26

**Table 2 molecules-28-02426-t002:** The relative RNA expression of lymphocyte IFN-γ, IL-2, and IL-4 contents with *p*-value in each group in synergistic stimulation of Se-MOP (0.195, 0.39, 0.78, 1.56, 3.125, 6.25, 12.5, 25 μg/mL) compared with PHA (10 μg/mL) after 48 h. Data from three separate experiments are expressed as means ± SD in triplicate.

Se-MOP μg/mL vs. PHA	IFN-γ		IL-2		IL-4	
	Mean ± SD	*p*-Value	Mean ± SD	*p*-Value	Mean ± SD	*p*-Value
0.195 vs. PHA	2.40 ± 0.07	0.0754	1.49 ± 0.01	0.0489	1.03 ± 0.07	0.9938
0.39 vs. PHA	2.40 ± 0.11	0.0751	1.68 ± 0.07	0.0047	1.72 ± 0.06	<0.0001
0.78 vs. PHA	3.93 ± 1.66	0.0002	2.54 ± 0.16	<0.0001	7.42 ± 0.14	<0.0001
1.56 vs. PHA	5.31 ± 0.10	<0.0001	3.75 ± 0.57	<0.0001	2.09 ± 0.08	<0.0001
3.125 vs. PHA	3.87 ± 0.84	0.0002	2.21 ± 0.06	<0.0001	1.95 ± 0.07	<0.0001
6.25 vs. PHA	2.68 ± 0.14	0.0249	1.85 ± 0.10	0.0005	1.76 ± 0.04	<0.0001
12.5 vs. PHA	2.29 ± 0.06	0.1131	1.62 ± 0.10	0.0093	1.16 ± 0.04	0.0909
25 vs. PHA	1.41 ± 0.07	0.9552	1.50 ± 0.01	0.0463	1.04 ± 0.05	0.9740

## Data Availability

Not applicable.

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
