# Peer review of "Selenium Nanoparticles Based on Morinda officinalis Polysaccharides: Characterization, Anti-Cancer Activities, and Immune-Enhancing Activities Evaluation In Vitro"

_molecules, 2023, doi:10.3390/molecules28062426_

Round 1

Reviewer 1 Report

Dear colleagues,
In this manuscript, the authors demonstrate influence of Selenium nanoparticles based on Morinda officinalis polysaccharides with characterization, anti-cancer activities, and immune-enhancing activities evaluation. The results are interesting. The figures reflect the results of the study. Despite the very good impression of the article, there are some remarks which could improve the article in my opinion, partly:

Checking for numerous missed space before source regarding;

Questionable order of M&M part after discussion;

Possible limitations for research should be described;

Conclusion could be added with digital parameter for stimulation of Se-MOP, increased immune capacity and improved immune by increasing  the expression of cytokines

In summary, I have been satisfied with the high level of the article. I believe this manuscript will attract significant attention from the research community. In my personal opinion, the article is very valuable, a great prospect for further research, and, after minor corrections, can be recommended for publication. 

Reviewer 2 Report

The article needs a very serious revision and cannot be accepted for publication in its current form.

- Legends for figures 1-4 are not very meaningful. Should be more detailed.

- The English language must be substantially revised and verified by a native speaker. showed most signific inhibition - must have shown the most significant inhibition.

- There are many phrases in the text in which there are no spaces between words.

- The work uses incubation at completely different time intervals. Either it is 48 and 72 hours, or it is exclusively 24 hours. However, the reasons for such different experimental approaches are not explained.

- Photographs of the cell cultures used for the experiments must be submitted.

- The characterization of the Se-MOP using an arsenal of physical methods is quite suitable for the present work. However, MOP particles should also be presented for comparison.

- 4.4 Selenium content determination. Despite the presence of references, the method should be briefly described.

- In this study, we discovered that nano-selenium was successfully dispersed with the help of Morinda officinalis polysaccharide. After combining with nano-selenium, the activity of Morinda officinalis polysaccharide was significantly enhanced. Not all experiments present data in comparison to MOP. In addition, MOP has a spectrum of antioxidant properties. It is necessary to carry out experiments with a source of selenium in the form of pure selenium nanoparticles (SeNPs).

- Discussion frankly weak. I would recommend the authors to get acquainted with the work of the team that work with selenium nanoparticles. https://pubmed.ncbi.nlm.nih.gov/?term=turovsky+e&filter=years.2022-2022&sort=date

- The output is very detailed. Much attention has been given to the production and characterization of Se-MOPs, while their physiological effects have received less attention. The conclusion is rather speculative as I have not seen experiments with a source of selenium without Morinda officinalis polysaccharides.

Round 2

Reviewer 2 Report

The authors have done a serious work on the article. All my comments were taken into account. I believe that the article can be accepted for publication in its current form